# Defining acceptable data collection and reuse standards for queer artificial intelligence research in mental health: protocol for the online PARQAIR-MH Delphi study

Dan W Joyce [ID],[1] Andrey Kormilitzin,[2] Julia Hamer-Hunt,[3] Kevin R McKee [ID],[4] Nenad Tomasev[4]

¹Department of Primary Care and Mental Health and the Civic Health Information Laboratory, University of Liverpool, Liverpool, UK
²Department of Psychiatry, Oxford University, Oxford, UK
³Department of Psychiatry, University of Oxford, Oxford, UK
⁴DeepMind, London, UK

**Correspondence to**
Dan W Joyce;
d.joyce@liverpool.ac.uk

## ABSTRACT

**Introduction** For artificial intelligence (AI) to help improve mental healthcare, the design of data-driven technologies needs to be fair, safe, and inclusive. Participatory design can play a critical role in empowering marginalised communities to take an active role in constructing research agendas and outputs. Given the unmet needs of the LGBTQI+ (Lesbian, Gay, Bisexual, Transgender, Queer and Intersex) community in mental healthcare, there is a pressing need for participatory research to include a range of diverse queer perspectives on issues of data collection and use (in routine clinical care as well as for research) as well as AI design. Here we propose a protocol for a Delphi consensus process for the development of PARticipatory Queer AI Research for Mental Health (PARQAIR-MH) practices, aimed at informing digital health practices and policy.

**Methods and analysis** The development of PARQAIR-MH is comprised of four stages. In stage 1, a review of recent literature and fact-finding consultation with stakeholder organisations will be conducted to define a terms-of-reference for stage 2, the Delphi process. Our Delphi process consists of three rounds, where the first two rounds will iterate and identify items to be included in the final Delphi survey for consensus ratings. Stage 3 consists of consensus meetings to review and aggregate the Delphi survey responses, leading to stage 4 where we will produce a reusable toolkit to facilitate participatory development of future bespoke LGBTQI+–adapted data collection, harmonisation, and use for data-driven AI applications specifically in mental healthcare settings.

**Ethics and dissemination** PARQAIR-MH aims to deliver a toolkit that will help to ensure that the specific needs of LGBTQI+ communities are accounted for in mental health applications of data-driven technologies. The study is expected to run from June 2024 through January 2025, with the final outputs delivered in mid-2025. Participants in the Delphi process will be recruited by snowball and opportunistic sampling via professional networks and social media (but not by direct approach to healthcare service users, patients, specific clinical services, or via clinicians' caseloads). Participants will not be required to share personal narratives and experiences of healthcare or treatment for any condition. Before agreeing to participate, people will be given information about the issues considered to be in-scope for the Delphi (eg, developing best practices and methods for collecting and harmonising sensitive characteristics data; developing guidelines for data use/reuse) alongside specific risks of unintended harm from participating that can be reasonably anticipated. Outputs will be made available in open-access peer-reviewed publications, blogs, social media, and on a dedicated project website for future reuse.

## STRENGTHS AND LIMITATIONS OF THIS STUDY

⇒ This Delphi study examines the intersection of data science, artificial intelligence, and mental health-care for LGBTQI+ communities advancing on similar research that has focused on healthcare or sexual health.

⇒ Delphi studies enable a participatory approach to the development of consensus recommendations and guidelines.

⇒ The Delphi study will be led by a team from the UK, which may limit the generalisability of Delphi outputs to regions with similar societal attitudes and legislative mechanisms that protect the rights of LGBTQI+ people.

## BACKGROUND

Artificial intelligence (AI), machine learning (ML), and data-driven technologies are expected to deliver novel ways of understanding and improving mental healthcare.[1] In healthcare applications of AI/ML generally, there has been increased focus on the potential for unintended harm arising from biases present in data[2] and resulting from model assumptions. Two striking examples being racial biases in an algorithm deployed to identify increased healthcare needs[3] and commonly used models for estimating renal function (employing standard biostatistical

methods) have been shown to be poorly calibrated for estimating kidney disease in people of colour.[4]

The ambition of any data-driven learning health system[5] is to improve the care provided to patients by adapting provision to their specific needs. In the context of mental healthcare, LGBTQI+ (Lesbian, Gay, Bisexual, Transgender, Queer and Intersex) communities are known to have specific difficulties arising from *minority stress*[6 7] including victimisation, internalised prejudice, and isolation. Consequently, LGBTQI+ people experience higher rates of suicidal distress,[8] self-harm and suicide,[9] and differential lifetime prevalence of the most common mental disorders as a function of sexual orientation and gender identity (SOGI), ethnicity, and race.[10] National survey data support these studies, showing that for example, 3% of gay and bisexual men (compared with 0.4% of men in the general UK population) attempted to end their life by suicide in 2013[11]; over 80% of trans-identifying young people have self-harmed at some point in their lives compared with around 10% in the general population[12] and 24% had accessed mental health services[13] in the preceding 12 months.

We note that there is variation in cultural and societal definitions of 'mental health' and 'mental illness',[14] including the egregious assumption that LGBTQI+ identity is, by definition, a 'mental illness'.[15 16] In this Delphi process, while we include the biomedical definition of mental illness/disorder, we will use an inclusive and broad term—'mental distress'—defined as a constellation of experiences that cause distress for the person, result in a loss of social, personal or occupational function, and/or reduction in quality of life. Furthermore, in the proposed Delphi study, mental distress is something for which the individual would seek assistance from an external source (eg, from healthcare professionals or peer/community support), or where other stakeholders identify an unmet need (eg, an LGBTQI+ support community identifying lack of support for a specific set of problems in people who remain 'invisible' to healthcare services).

## Data quality

Supporting LGBTQI+, people requires high-fidelity data.[17 18] However, such data are ostensibly lacking for reasons including the following:

► A lack of harmonisation for the recording of SOGI data resulting in fragmented, incompatible data.[19 20]
► Poor recording rates for local data collection, beyond services focused on, for example, cis-gendered gay men and sexual health[11]
► Disclosure of SOGI characteristics to healthcare professionals is low, because LGB people experience healthcare organisations and professionals as threatening[21] and there is evidence that an individual's medical history, immigration status, level of internalised homophobia, and degree of connectedness to the LGBTQI+ community are significant factors for disclosure with bisexual men and women being the

least likely to disclose SOGI characteristics to healthcare professionals[22]
► Discrepancy between patient and healthcare professionals expectations around offending people by asking about SOGI characteristics, resulting in, for example, 80% healthcare professionals believing they may offend by asking about SOGI characteristics compared with 11% of patients reporting likelihood of offence.[23]
► Accessing healthcare is difficult for LGBTQI+ people; for example, in the UK's LGBT National Survey, 72% of people who had tried to access mental healthcare (24% of respondents had tried) described it was 'not easy'.[13]

## Appropriate data use

The straightforward imperative that we *require* better data collection is well documented,[24–26] but difficult to implement. Furthermore, there is less evidence on the specific and acceptable uses of data and explainable AI/ML technology to advance the provision of care for the LGBTQI+ community[27–29] and this problem pervades healthcare data reuse more generally. For example, a recent piece of investigative journalism on the UK Biobank claimed that 'Sensitive health information donated for medical research by half a million UK citizens has been shared with insurance companies despite a pledge that it would not be'.[30] In response, UK Biobank responded robustly,[31] arguing their stewardship of the data meant that 'Researchers from insurance companies are treated like all other commercial or academic researchers' and that the examples cited in reference[30] all had 'met the required tests of involving suitably qualified researchers and being health-related research in the public interest'. Biobank's current patient information leaflet[32] under the section 'Who will be able to use my information and samples?' explicitly states that 'Insurance companies and employers will not be given any individual's information, samples or test results'. Biobank participants might understand this to mean that insurance companies with direct commercial interest in decisions about them will never be given their individual data for the purpose of, for example, assessing their insurance liability or risk. However, because insurance companies can be considered suitably qualified commercial researchers, participants might hold different opinions on *any* use of data in Biobank for purposes linked to the insurance industry on the grounds it cannot be health-related research in the public interest.

This example illuminates relevant themes for LGBTQI+ communities—namely the need to understand:

► How SOGI data can be *meaningfully collected, stored, and processed* in a way that is compatible with the language and norms defined by LGBTQI+ communities.
► The *acceptable use-cases* for using individual and population level SOGI data collected in routine clinical care.

This paper describes a protocol for a Delphi process to develop a consensus on these questions.

## Rationale for a participatory approach

Patient, public, and stakeholder involvement in mental health research has an established history and is motivated by Boivin et al[33] stakeholder involvement as an ethical imperative with the expectation that this may improve the quality, relevance, and uptake of research.[34] Arnstein's 'ladder of citizen participation'[35] is often cited as an anchoring principle for meaningful stakeholder involvement and participatory design[36] with contemporary definitions[37] defining patient and public involvement (PPI) as for example, 'a process whereby professionals and those traditionally on the receiving end of their 'expertise' (eg, patients/service users/marginalised citizens) can collaborate with the goal of achieving outcomes that arguably cannot be achieved otherwise. It should engage the talents and experience of all involved and support the egalitarian relations and conditions needed to make the most of them'. In healthcare, the defining summary statement is 'no decision about me, without me'[38 39] and adopting this principle of empowerment and codesign for healthcare AI comes with unique challenges.[40] Participatory approaches present a necessary step in the safe development of AI systems for delivering positive impact[41] and participatory design can play a critical role in empowering marginalised communities to take an active role in constructing research agendas and outputs; for example, in applications spanning architecture, the environment and planning,[42 43] community building[44] and education.[45]

A central tenet of AI research applied to healthcare should be that affected communities are active participants in the codesign and production of services and technologies to avoid (usually) unintended harms, to mitigate unforeseen consequences of technical processes and the avoidance of sociotechnical 'blind spots'. In the application of AI specifically to LGBTQI+–inclusive mental healthcare, the interaction of minority stress[6] with the stigmatisation of mental illness more generally[46] presents a quagmire of acceptability, safety, and healthcare equity concerns. We argue that these can only be addressed through a participatory process that identifies how services and technologies understand, collect, codify, and use the communities' data to ensure they benefit. In health sciences, the Delphi technique has been useful for establishing a consensus on 'complex issues where knowledge is uncertain and incomplete'[47] and where evidence synthesis from, for example, experimental or epidemiological data is difficult.[48] Consistent with our aims for PARticipatory Queer AI Research for Mental Health (PARQAIR-MH), the method can enable a diversity of perspectives to be represented during consensus development.

## Aims of the Delphi study

The application of data-driven technologies to high stake applications—such as healthcare—requires high fidelity, comprehensive and therefore sensitive data to help mitigate biases, improve fairness, prevent inequality, and to ensure representation. Consequently, stewards and guardians of such highly granular data must describe (as unambiguously as possible) the parameters on who will use this data and for what purpose. The aims of this Delphi study are therefore:

1. To establish consensus on how to collect, code, and harmonise SOGI data in the context of improving provision of mental healthcare.
2. Assuming high-fidelity SOGI data are collected and available, to establish consensus on the scenarios and use-cases for acceptable reuse of this data in data-driven technologies (eg, AI, ML, population health, and epidemiology)—including identifying use-cases that (according to the community stakeholders) constitute absolute 'hard no' and qualified 'potentially yes' cases.

The study will deliver the following outputs:

1. A 'best practice' toolkit, defined by LGBTQI+ community stakeholders, for AI developers, data scientists, and healthcare institutions to implement when collecting and recording SOGI characteristics for the people they serve. While this toolkit will be developed specifically for mental healthcare, some insights may be informative for other health and social care contexts.
2. An online 'playbook' describing concrete example scenarios of SOGI data use that are clearly unacceptable or that may be acceptable with qualifications or with specific safeguards and conditions
3. Open-access academic paper(s) that summarise the outcomes of the Delphi study, directing stakeholders (policy makers, institutions, and teams/individuals) on best-practice for using data-driven technology in the context of LGBTQI+ people and mental health.

Our focus for the Delphi study will be on data that are expected to be collected routinely and in clinical or health settings (whether public, private, or third-sector providers). Therefore, we will not consider the reuse of data from, for example, social media sources, blogs, or other self-publishing platforms. Factors that explicitly address the most appropriate models of healthcare service design and delivery,[13 49] while certainly relevant for people's experiences and future engagement with providers, will be out-of-scope for PARQAIR-MH due to the specific focus on ways to use data to improve LGBTQI+ affirmative care.

## METHODS/DESIGN

The multistage consensus method will follow recommendations for the Delphi technique.[50] The fundamental principles of Delphi approaches are to exploit the 'wisdom of crowds' (multiple experts), to collect anonymous feedback, and to iterate over multiple-rounds.[51] These principles remain a constant feature of Delphi studies but the method has been applied to (and modified to account for) different objectives (eg, policy issues

and decision-making[52]), applications (eg, healthcare research[53–55]), and mechanisms of executing the Delphi process—notably, the adaptation of the traditional Delphi to online-based platforms.[56]

The Delphi process comprises multiple stages, is overseen by an executive committee (the authors of this protocol), and an advisory working group (composed of representative stakeholders). In outline (see figure 1), the stages are:

1. Conduct literature review, recruit advisory working group, and define terms-of-reference (ToR).
2. Advisory and executive working groups collectively define the first questionnaire for the Delphi rounds; simultaneously, the advisory working group and executive group will advertise and manage recruitment of the survey group.
3. Three sequential Delphi rounds are completed anonymously by the survey group participants via a secure web-based online platform.
4. Defining the final consensus on the outputs of the Delphi rounds and a final consensus meeting with the executive, advisory working and survey groups.
5. The executive and advisory working group then build and deliver outputs (web-based toolkit, guidance including concrete example scenarios, and open-access papers summarising findings).

### Patient and public involvement statement

Coauthor Julia Hamer-Hunt, a lived-experience practitioner, consulted on the principles and design of the Delphi process from conception to the final draft of this protocol. The Executive Committee will be assembled by a targeted approach through the authors' professional networks, alongside open calls on social media, to ensure a diverse, equitable, inclusive, and representative panel of stakeholders (including patients and public) to oversee the Working Group and execution of the Delphi consensus process.

### Working groups: composition and recruitment

The PARQAIR-MH) working group will include the following:

1. An *executive group* responsible for the overall execution of the project, organisational/operational processes to conduct, disseminate, and report on the Delphi process. This group will consist of the authors of this manuscript.
2. An *advisory working group* who will lead the final-stage consensus meeting and be drawn from experts from the AI/ML, ethics, health policy, mental health professionals, and PPI stakeholder groups. We will aim to recruit 10 people to the advisory group.
3. A *survey group* of people (with similar composition to the advisory group) who will participate in the Delphi survey. This online survey group will be open to any interested (self-selecting) stakeholders able to provide informed consent and able to access the online survey. Our aim is to recruit a minimum of 50 participants to

meet the heuristic of requiring approximately 30–50 participants.[57–60] Of note, we expect attrition over the three Delphi rounds, but as consensus requires participants to complete all stages (and these will be conducted synchronously, with everyone asked to complete rounds in a certain time-period before the study progresses to the next round), we will not recruit additional participants to account for those leaving the study after only completing one or two rounds.

The advisory and survey groups will be composed of the following:

1. People with lived experience of mental health service use from the LGBTQI+ communities.
2. People representing charities, NGOs (non-governmental organisations), and other campaigning groups focusing on the mental health of people from the LGBTQI+ community.
3. Domain experts drawn from mental health professionals, invited to participate from national and international LGBTQI+ communities to include people with lived experience of mental health problems.

Recruitment for the advisory working group will be via snowball and opportunistic sampling using the executive committee's professional networks (spanning mental healthcare, PPI, science/engineering, charities, and support networks for LGBTQI+ people in the community and technology industry):

► Directly approaching community groups and charities supporting LGBTQI+ people with an interest in mental health.
► Directly approaching LGBTQI+ policy leads in the UK's National Health Service (NHS) and the Royal College of Psychiatrists.
► By arranging an online 'town hall' event, announcing the PARQAIR-MH initiative with publicity on public social media (X/Twitter) platforms and closed community platforms (eg, Queer in AI) to publicise the initiative and invite participation.

Recruitment for the survey group will be conducted similar to the advisory group, but in addition, we will request support in cascading publicity/advertising for participation in the survey group (those completing the three Delphi rounds) to people known to or using charity/community groups and again, using colleagues in the executive and advisory group's respective professional networks.

Participants will not be financially compensated for their contributions, but with their consent will be given attribution on the project's website and acknowledged in academic publications. People volunteering for the advisory working group will be offered coauthorship on academic publications.

A particular challenge with online Delphi studies is that participants will be self-selecting, and it is difficult to achieve appropriate representation, for example, across SOGI as well as different stakeholder sectors. We believe that it is unethical to ask volunteers to describe their SOGI characteristics in order to selectively invite people

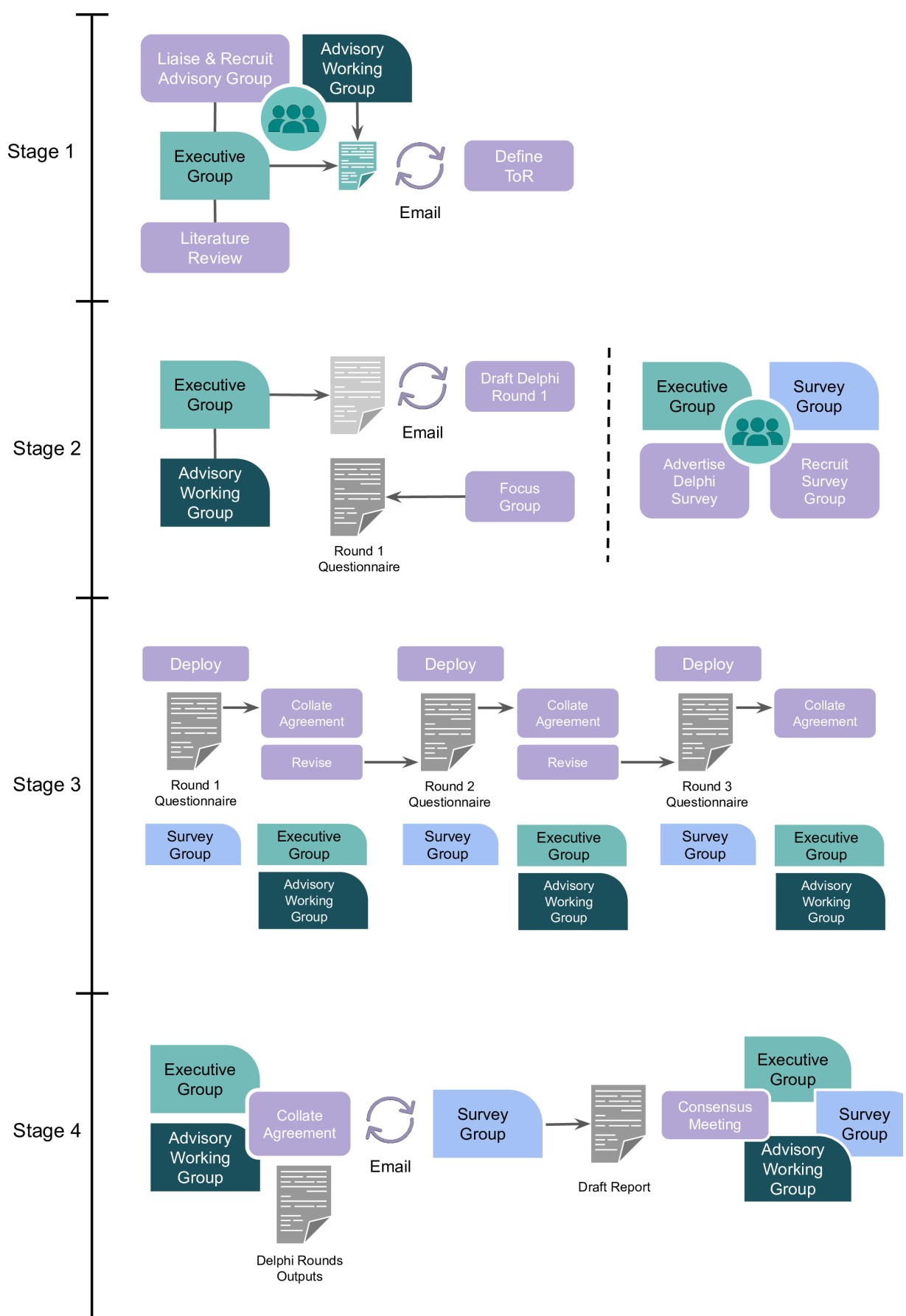

**Figure 1** Stages of PARQAIR-MH Delphi study. PARQAIR-MH, PARticipatory Queer AI Research for Mental Health; ToR, terms-of-reference.

to ensure diversity and representation in these groups. We acknowledge that this may limit the representativeness of these groups; we will instead describe the groups' composition and report any impacts this has on the conclusions and generalisability of findings.

## Stage 1: defining the terms-of-reference and literature review

To focus the initial round of the Delphi questionnaire, the executive group will review existing literature to identify:

► Existing healthcare guidelines for the collection of SOGI data for LGBTQI+ people.
► Studies of perceptions, attitudes, and experiences to disclosure of SOGI data in healthcare settings for LGBTQI+ people.
► Example applications of data-driven technology (in particular, AI) in LGBTQI+ mental health support to include those that expose benefits, risks, and harms specific to that community.

This review will include both published, peer-reviewed academic literature, governmental, NGO, and charity surveys as well as publicly available policy documents. Special attention will be given to surveying the ethics and fairness literature, to identify promising approaches for ensuring privacy and safety of AI systems. The glaring absence of analyses of disparate impact of AI on queer communities[61] further justifies the need for a deeper community involvement.

Following a review of the literature, the executive committee will produce a summary of the findings alongside a draft ToR, describing the Delphi study's aims, scope and intended deliverables and outputs. The ToR will be circulated via email for comments and revision by the advisory working group over a period of 4 weeks to agree on the final ToR.

The executive group will conduct a targeted literature review of existing literature, guidelines, and toolkits that inform the aims of this Delphi study and will form the preparatory step for the initial Delphi questionnaire. There are two primary foci for the Delphi study that require a review of literature and other research outputs: (a) how to capture SOGI data (so it is complete, valid, and collected in an affirming way) and (b) the parameters of this data's reuse (similar to responsible data stewardship for AI and data-driven technologies[62]).

Following the framework introduced by Arksey and O'Malley,[63] we will search relevant databases (eg, PubMed and Crossref) for papers and guidance documents published from 2000 to 2024. For the review of data-collection practices, we will employ a combination of controlled vocabulary terms and keywords related to LGBTQI+ communities (eg, 'sexual and gender minorities', 'LGBT', and 'LGBTQI+') and routine data collection in healthcare-related domains (eg, 'routine data', 'electronic patient', and 'electronic health'). Our initial reviews (eg, described in this protocol paper) revealed that case-studies describing existing practices, guidelines ('playbooks'), and toolkits are often

not part of the traditional scientific literature and in particular, web-based resources are often less visible as academic outputs. For this reason, we will perform web-searches with similar terms and additionally search websites of relevant organisations (eg, the WHO and the American Medical Association) for potentially relevant guidance documents.

For the aim of demarcating the parameters of acceptable data-reuse, we will augment the controlled vocabulary terms related to LGBTQI+ communities with terms to capture scenarios, permissible, and unacceptable use-cases and search for publicly available impact assessments relevant to queer-affirming healthcare. Our initial searches suggest a majority of scholarly activity describes existing, or sometimes predicted, harms from, for example, facial recognition technology.[64 65] We will need to develop anticipated use-cases and scenarios that might predictably arise in the application of data-driven technology in mental healthcare (or healthcare more generally) which are sparse—a recent exception being the application of conversational AI to provide mental health support for the LGBTQI+ community.[66]

For both targeted reviews, two reviewers from the executive group will independently screen titles/abstracts (for the research literature) or 'executive summaries' or landing pages (for web-based resources) to identify eligible artefacts for full review and inclusion. Reviewers will first attempt to resolve disagreements through discussion; if discussion fails to resolve disagreements, a third reviewer will break ties. Subsequently, we will review the full text of eligible documents and artefacts for insights and claims relevant to the two topics.

## Stage 2: Initial questionnaire definition

With an agreed ToR—and drawing on examples of existing practice and assets arising from the literature search—the executive group will draft an initial questionnaire for the first round of the Delphi process. In parallel to the drafting of the first questionnaire, the executive group will advertise the online Delphi questionnaire study as described above (Working Groups: Composition and Recruitment).

It is anticipated (subject to the literature review and input from stakeholders in the advisory group) that the content of the first draft questionnaire will cover the following topics, aligned with the study aims:

1. LGBTQI+ community preferences for collecting, recording, and harmonising SOGI data.
2. The parameters for the acceptable (re)use of SOGI data for improving healthcare systems to include the following example use-cases:
   – The use of automation (eg, AI-driven chatbots or recommender systems)
   – Decision support (eg, identifying risk factors for individual people)

– Configuring/commissioning services (eg, auditing SOGI data for adapting existing, or developing new, services).

For the initial (and subsequent) questionnaires, it is anticipated that data collected will be structured (and consensus defined) as follows:

► Some items will invite participants to provide a two-alternative forced choice (eg, 'Would you prefer to provide information on your sexual orientation by (A) selecting a label that encompasses both attraction and partnering (eg, heterosexual, gay, and lesbian), or (B) providing separate information on your attraction and partnering preferences?'). For these items, a consensus will be defined as when ≥70% of participants respond with the same answer.

► Where questions invite an ordinal, positive-, or negative-preference response, participants will be asked to provide an answer on a 7-point Likert scale (eg, 'A clinical service wishes to use it's patient's self-described gender identity data to report on the demographics of the service; Is this an acceptable reuse case?') with anchors 'Strongly Disagree', 'Strongly Agree', and 'Neutral' coded as 1, 7, and 4, respectively. For these items, consensus will be defined as ≥70% of participants responding with 'Agree' or 'Strongly Agree' (positive consensus) or, 'Disagree' or 'Strongly Disagree' (negative consensus)

► Some items will present a longer form scenario, followed by a number of related questions that invite two-alternative and/or ordinal preference responses; in addition, if there is scope for nuance or a need for narrative description of why a particular answer was given, free-text fields will be available for additional comments. Narrative responses will be summarised and presented, for example, as 'qualifications' to the topics described in the question.

For the first round, the questionnaire will conclude with a free-text invitation to suggest areas, topics, scenarios, or specific questions/items that the participant feels where neglected and this will be taken into account for the design of round 2.

## Stage 3: Delphi rounds

The Delphi process will consist of three sequential rounds, all conducted via a web-based questionnaire delivery platform. Each participant will be identified only by an email address (that they provide on starting the first questionnaire round). IP addresses will not be retained or used to identify participants or their survey responses. Each email address will be assigned a unique participant number in a participant table to ensure that the same participants are responding to each of the three rounds and so that invites for subsequent rounds can be distributed to those completing the first round. The participant table will be retained securely and available only to the executive group.

The three rounds are as follows:

► *Round 1*: participants will be presented with direct questions or short vignettes describing either existing (from the literature review) or hypothetical use-cases for SOGI data collection and reuse. Participants will be given a fixed time period within which to complete the questionnaire round. At the end of the period, the executive group will retrieve responses from the web-platform and store them securely, identifying participants by their unique participant number and separately storing the participants email address. Agreement on each question will be conducted as described above (stage 2: Initial Questionnaire Definition) alongside analysis of any narrative responses. The resulting questions, responses, and agreement will be summarised by the executive group and presented to the advisory group in an online meeting that decides which questions/items are to be retained, modified, or ejected from the subsequent round. For example, items with a clear consensus will be removed from round 2, whereas items that fail to produce consistent responses will be modified.

► *Round 2*: the round 1 participants will be notified via email from the executive group asking them to participate in the second questionnaire round. This second round will be prefaced with an anonymised summary of the round 1 responses, including indicating which items were subsequently removed due to consensus being reached. As for round 1, at the end of a defined time-period, the round will be ended, data retrieved and analysed for consensus and revision of the questionnaire for the final round.

► *Round 3*: the final round will follow the same process as previous rounds. However, we expect the final round to contain items addressing topics which remain particularly contentious (ie, where agreement between participant responses remains low). Participants will be made aware that in the final round, any items that do not reach consensus will be reported as areas with uncertain conclusions and they will be reported as such; this is to ensure participants are aware that if they recognise items on similar topics/themes from previous rounds, they should not necessarily modify their responses purely because this represents the final round.

## Stage 4: Consensus process

The executive group will collate the rounds of questionnaires, providing a summary of the questions and the corresponding numerical measure of agreement among participants. Attention will be paid to highlighting areas where there remained lack of agreement after three rounds. The advisory group will be consulted via email to enable revision on the summary report before being emailed to all survey group participants who will also be invited to reply with commentary on the report.

A consensus meeting will be advertised to participants in the advisory and survey groups, inviting them to attend and discuss proposals for how the summary report can be

presented as outputs to meet the aims of the study. This consensus meeting will be online using a video conference platform and participants at the meeting will be asked to use a pseudonym screen name, and to keep their video feed switched off (ie, audio only) to help preserve anonymity. The executive group will organise and moderate this meeting, with one member (an experienced social scientist) designated a non-voting chair. In addition, we will invite stakeholders using the executive and advisory group's professional networks. We expect the consensus meeting will therefore have representation from:

▶ NHS and University PPI/Engagement groups.
▶ Stakeholders from the survey group (rounds 1–3 participants).
▶ Clinicians working in the mental health sector.
▶ Ethicists.
▶ AI researchers and data scientists.
▶ NGOs and charity stakeholders for LGBTQI+ mental health.

Importantly, we recognise that consensus may be difficult to achieve for certain topics and themes; for example, some participants might have a strong opinion that the use of automation and data-driven technologies is unacceptable in any aspect of mental healthcare delivery. Given the complexities of defining consensus,[67] themes where agreement could not be reached will be reported and highlighted in the final outputs (eg, the web-based toolkit, playbook, and in academic publications).

### Outcomes and dissemination

At the consensus meeting, the outputs deemed necessary and sufficient for a toolkit will be discussed; for example, the format and medium for the researcher 'checklists', guidance, and 'playbook' documents (describing scenarios and offering advice on acceptability according to the outputs of the Delphi questionnaires and the consensus meeting's recommendations) that we expect to take the form of a recommendations white paper and case-study format similar to prior work in related areas.[62 68 69] Following this, the executive group will invite the advisory group to contribute to writing a summative report for submission to an open-access, peer-review journal.

The key outputs (toolkits, guidance documents, and advice for replicating the Delphi process) and findings (including open-access, peer-reviewed papers) will be made available on a website (similar to the equator network, https://www.equator-network.org/) that will be maintained by the executive committee. The aim is to provide a participatory design-inspired open and transparent process for communities and organisations to either deploy the consensus and toolkit in their own localities, or to replicate the process to derive locally informed versions of the toolkit/consensus.

Stakeholder involvement in all outputs from the proposed Delphi process will be transparent and explicitly described, including composition of the Executive Committee and Working Group. Specific PPI will be reported using the GRIPP2[70] reporting guidelines.

## DISCUSSION

### Scope and generality

Existing work on SOGI data collection and harmonisation reflects a largely Western geographical focus including the European Union, UK, and USA.[17–19] The pending UN Report to the Human Rights Council[71] on SOGI emphasises healthcare equity for LGBTQI+ communities (including data collection/harmonisation as a key enabler) while previous UN mandate reports[72] acknowledge under-representation from regions of the world with hetero-normative cultural attitudes or where people from LGTBQ+ communities are persecuted. Similarly, different societies and cultures' formulation of mental illness in terms of aetiology, stigma, implications for individuals, family, and wider society vary to the extent that a dominantly Western biomedical model (ie, proposed to emphasise the individual as the locus of mental illness and disorder) is seen as unhelpful (see reference[14] for a review). While the overarching PARQAIR-MH process remains general, the outcome of its initial application in the UK will be limited and localised in its immediate practical utility, necessitating replication studies.

### Limitations

The patient and public perception of clinical applications of AI is relatively under-studied; one systematic review[73] of 23 mixed-methods studies found no studies specifically addressing mental healthcare. The review exposed some polarisation around themes of accountability (of a decision made using AI), concern around 'boundary cases' (ie, rare diseases or uncommon situations), and a divide around risk of worsening or improving healthcare outcomes, equity, and justice. Importantly, they note that the perspectives of under-represented groups were rarely included or studied in the sampled literature.

Given this, we expect similar polarity in our Delphi process which may limit the extent to which consensus can be reached. Consequently, we will report separately on subsets of items achieving consensus, those where no consensus could be reached and a clear description of contentions arising in both subsets.

### Protocol reuse and utility

Considering the rising need for a wider community involvement in AI design, and this being one of the very first AI participatory studies designed specifically for the LGBTQI+ population, we hope that the proposed protocol will help inform a multitude of future participatory research directions. Indeed, the issues of data collection, data use, fairness and safety, are central to AI development across mental healthcare, healthcare, as well as numerous other domains and use cases.

Consistent with the central tenets of participatory design, this protocol needs to be applied locally to capture the local variation in perspectives, needs, and healthcare systems. Repeated application of the protocol may result in different consensus statements, reflecting these local differences. We would therefore strongly encourage

worldwide replication studies, complementing the initial study planned in the UK. In terms of utility, PARQA-IR-MH aims to help inform digital health policy and the design of inclusive mental healthcare technologies going forward.

## Ethics and dissemination

Participants in the Delphi process will be recruited by snowball and opportunistic sampling via professional networks and social media (but not by direct approach to healthcare service users, patients, specific clinical services, or via clinicians' caseloads). Participants in the survey group will not be required to share personal narratives and experiences of healthcare or treatment for any condition. The Delphi rounds will be completed online, asynchronously (as participants may be in different time zones) and pseudonymously using a web-based, secure platform hosted at the University of Liverpool. Participants will be required to provide informed consent (via an online form), after reading a participant information sheet describing the issues considered to be in-scope for the Delphi (eg, developing best practices and methods for collecting and harmonising sensitive characteristics data; developing guidelines for data use/reuse), an outline of the risks of unintended distress arising from participation (in so far as this can be reasonably anticipated) and informing participants of the options to withdraw and remove their data from the study. After each Delphi round, participants will be offered the opportunity to participate in an online debriefing session. Participants volunteering to assist in the final consensus process (to agree the final output of the Delphi rounds) will be asked to participate in the online video-conference pseudonymously (ie, audio-only, identifying themselves on-screen using a pseudonym). The study, consent processes, data protection, and participant-facing information materials have been approved by the University of Liverpool's Research Ethics Committee (REC Reference: 12413; 24 July 2023).

Outputs will be made available in open-a-cess peer-reviewed publications, blogs, social media, and on a dedicated project website for future reuse.

**Contributors** The study was conceived and designed by authors DWJ, AK, JH-H, KRM, and NT. The drafting and revising of this protocol paper was completed by DWJ, AK, JH-H, KRM, and NT. Final approval for the version published was agreed by DWJ, AK, JH-H, KRM, and NT who also agreed to be accountable for all aspects of the work in ensuring questions related to the accuracy or integrity of any part of the work are appropriately investigated and resolved.

**Funding** This research received no specific grant from any funding agency in the public, commercial, or not-for-profit sectors. Authors AK and DWJ gratefully acknowledge the National Institute for Health and Social Care Research (NIHR) (grant: AI_AWARD02183) for partial salary support relating to this work.

**Competing interests** DWJ and AK are partially supported by an NIHR grant (AI_AWARD02183) which explicitly examines the use of AI technology in mental healthcare provision. NT and KRM are employees of Google DeepMind, an AI research company. JH-H has no competing interests to declare.

**Patient and public involvement** Patients and/or the public were involved in the design, or conduct, or reporting, or dissemination plans of this research. Refer to the Methods section for further details.

**Patient consent for publication** Not applicable.

**Provenance and peer review** Not commissioned; externally peer reviewed.

**ORCID iDs**
Dan W Joyce http://orcid.org/0000-0002-9433-5340
Kevin R McKee http://orcid.org/0000-0002-4412-1686

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
