## [Reviewer comments · BMJ Open]

ARTICLE DETAILS

TITLE (PROVISIONAL)	Defining Acceptable Data Collection and Re-use Standards for Queer Artificial Intelligence Research in Mental Health: Protocol for the Online PARQAIR-MH Delphi study
AUTHORS	Joyce, Dan; Kormilitzin, Andrey; Hamer-Hunt, Julia; McKee, Kevin; Tomasev, Nenad

VERSION 1 – REVIEW

REVIEWER	Kent, Lisa Queen's University Belfast, Centre for Public Health
REVIEW RETURNED	08-Nov-2023

GENERAL COMMENTS	Thank you for the opportunity to review this study protocol investigating an interesting topic. The study appears to be focused on participatory design in artificial intelligence research in mental health within the LGBTQI community. However, the potential results could also have an impact on areas of healthcare beyond mental health, and data-driven research beyond AI-focused research, for the benefit of the LGBTQI+ community. I note the following in relation to the editor's instructions for reviewers of study protocols: 1) The authors have not stated if the study is planned or currently ongoing, and dates are not included2) Since it is not clear whether the study is in fact ongoing, it may not be practical or possible to address some of the points raised within the review concerning methodology GENERAL COMMENTS Although the title of the study describes a tight remit, the protocol that follows appears to refer more broadly to collection and re-use of data pertaining to sexual orientation and gender identity within healthcare systems. The 3 primary domains outlined in the section on Focus (page 9) do not clearly map to the focus and methods outlined within each stage in the methods/design section. In particular, the methods do not seem to adequately address the second domain of barriers/obstacles to disclosure. The authors could also consider revisiting the points described on page 7 lines 41 to 50, to ensure that these also map to the focus and methods described. The protocol would greatly benefit from a clearly defined aim. It is difficult to see what the end result of this study should be – is the expected output to be a consensus statement for AI research, guidelines for data collection/re-use, or a toolkit for AI developers?
--

	In addition, each stage in the process might benefit from clearly stipulated objectives. BACKGROUND Page 7 line 24 - accessing healthcare is difficult for LGBTQI+ people, for example, 28% of people in the UK's LGBT National Survey described it was "not easy" to access mental healthcare (14) -How does this compare with the general population? METHODS/DESIGN The authors refer to a reference from 1975 to support the Delphi approach. Does the authors' approach deviate in any way from this early description of the Delphi approach, and if so, is there evidence to support choice of methods? The authors allude to using the snowball recruitment technique in the ethics and dissemination section, however they may wish to consider adding a section in the methods on recruitment of participants. This could include methods they intend to use to recruit not only those with lived experience but also clinicians, ethicists, researchers etc. The authors may also wish to describe the following in this section: 1) reimbursement and compensation being offered to participants (if any), 2) target recruitment numbers of participants from each stakeholder group, 3) further methods of recruitment that might be used if there is an imbalance in numbers between stakeholder groups, 4) any efforts that will be made to ensure a representative spread of SOGI characteristics within the LGBTQI+ group, 5) any efforts that will be made to record the geographical spread of participants (even within the UK, it is possible that different regions/nations might have different experiences), 6) whether additional participants will be recruited if there are high numbers of drop outs between stages/rounds, 7) will participants be provided with training. Literature review stage This section could benefit from more detail. Whilst it is appreciated that methods within this evidence synthesis stage were perhaps chosen for pragmatic reasons, it could be worth considering using a more systematic approach. As a suggestion, the authors may wish to consider rapid reviews or scoping reviews, depending on the objectives of this stage and the type of literature being reviewed. For example, further details on search strategies, study/publication selection criteria and data extraction might be useful to readers. In addition, each of the areas to be covered in the literature review appear to be separate concepts that could benefit from being considered thoroughly and independently. Will a separate review be conducted for each? Page 12 – line 11 – Is the creation and review of the Terms of Reference a separate stage requiring its own section in the protocol? How do the authors intend to facilitate the review? Will the participants be given criteria against which they are to critique the Terms of Reference? Focus group stage? Other research using a similar approach often includes a focus group stage conducted alongside the literature review in order to
--	--

	fill gaps in the evidence and add depth of understanding. The results of both the literature review and focus groups are then used to inform the first round of the Delphi process. Could the authors comment on their decision to not include focus groups at this stage? Delphi Stage How will the Delphi be conducted? In person vs online? Anonymous participants? Round 1 –Please provide more information on the categorical responses that will be presented to the participants, and how agreement will be judged? A minor point is that the stated outcome of the Delphi stage is “consensus statement”, however a subsequent stage is described specifically for agreeing the consensus statement. The authors might wish to consider rephrasing the outcome for the Delphi Stage as “draft consensus statement” or similar. Consensus meeting stage The authors refer to a focus group, however they might wish to avoid using this term as focus group research typically involves techniques that are not described here. Will an experienced facilitator / non-voting chair lead the consensus meeting? How many participants will you aim for to ensure meaningful engagement? Will results of previous stage be stratified by stakeholder group prior to being presented? Outcomes/Dissemination Stage Page 14 Line 50 – how will the results of the prior stages feed in to the white paper? What is your approach to developing the case-study. Is there some text missing from the end of this sentence after “e.g.”? Limitations A further limitation that the authors may wish to consider – participants may alter their response based to fit the majority view, particularly if there are perceived differences in seniority or expertise. Ethics Could the authors please consider adding a brief statement on whether ethical approval is required for this study? Is informed consent from participants required? If so, what materials do you intend to provide to facilitate this process?
--	---

REVIEWER	Harrington, Kyle
	University of Nottingham, Institute of Mental Health
REVIEW RETURNED	17-Nov-2023

GENERAL COMMENTS	The protocol document is of high quality, well-explained and well-motivated. The only two comments I have refer to the "data quality" section of the paper. Authors suggest there is a misalignment of patient and healthcare professionals expectations around data collection with 80% of healthcare professionals believing they may cause offense compared to 11% of patients reporting likelihood of offence. Given
--

	that a healthcare professional may come in to contact with numerous patients, is it really correct to say that the healthcare professionals have misaligned expectations? How much does the accessibility of healthcare for LGBTQI+ people differ from the population more generally? Could the authors provide context?
--	--

VERSION 1 – AUTHOR RESPONSE

Reviewer: 1

Dr. Lisa Kent, Queen's University Belfast

Comments to the Author:

Thank you for the opportunity to review this study protocol investigating an interesting topic. The study appears to be focused on participatory design in artificial intelligence research in mental health within the LGBTQI community. However, the potential results could also have an impact on areas of healthcare beyond mental health, and data-driven research beyond AI-focused research, for the benefit of the LGBTQI+ community.

I note the following in relation to the editor’s instructions for reviewers of study protocols:

- 1) The authors have not stated if the study is planned or currently ongoing, and dates are not included
- 2) Since it is not clear whether the study is in fact ongoing, it may not be practical or possible to address some of the points raised within the review concerning methodology

Response:

In the Ethics and Dissemination section, we have highlighted that the study is intended to run from June 2024 through to January 2025 (the Delphi component) with outputs expected in early 2025. Given the study has not yet begun, we have been able to incorporate the suggestions presented below, except where they would be outside parameters agreed as part of our research ethics approval.

GENERAL COMMENTS

Although the title of the study describes a tight remit, the protocol that follows appears to refer more broadly to collection and re-use of data pertaining to sexual orientation and gender identity within healthcare systems.

Response:

The Editor similarly requested that the title be refined to better describe the paper along style guidelines for the journal; as such, the title has been revised to reflect that the Delphi study will

include a broader remit to include collection and re-use of data around sexual orientation and gender identity. The new title is “Defining Acceptable Data Collection and Re-use Standards for Queer Artificial Intelligence Research in Mental Health: Protocol for the online PARQAIR-MH Delphi study”

The 3 primary domains outlined in the section on Focus (page 9) do not clearly map to the focus and methods outlined within each stage in the methods/design section. In particular, the methods do not seem to adequately address the second domain of barriers/obstacles to disclosure. The authors could also consider revisiting the points described on page 7 lines 41 to 50, to ensure that these also map to the focus and methods described.

Response:

Thank you for this observation – on reflection, the second focus (barriers to disclosure) is somewhat tangential to the other two (collecting and re-using data) and risks diluting the study’s findings. We have amended the study aims accordingly and added a clear example (motivating the second aim) on lines 106–128. Further, we noted repetition of the three themes, variously defined as aims and foci for the study, and harmonised them with a section for a clearly defined aim / outputs (see response below)

The protocol would greatly benefit from a clearly defined aim. It is difficult to see what the end result of this study should be – is the expected output to be a consensus statement for AI research, guidelines for data collection/re-use, or a toolkit for AI developers? In addition, each stage in the process might benefit from clearly stipulated objectives.

Response:

Our original manuscript contained a number of aims, objectives and deliverables distributed throughout the document. We have now harmonised these into one section “Aims of the Delphi Study” and included a clear set of related expected outputs which are referenced throughout the manuscript. Please refer to lines 164 to 190 in the revised manuscript.

BACKGROUND

Page 7 line 24 - accessing healthcare is difficult for LGBTQI+ people, for example, 28% of people in the UK’s LGBT National Survey described it was “not easy” to access mental healthcare (14)

-How does this compare with the general population?

Response:

We are not aware of any direct comparator data that a) used a similar community sample and b) that speaks directly to the question of difficulty of access. Various CQC surveys (UK based) report satisfaction results with services (e.g. the 2022 mental community mental health survey <https://www.cqc.org.uk/publications/surveys/community-mental-health-survey>) but only report summaries such as “40% of people had ‘definitely’ seen services enough for their needs”. Given the

absence of data with comparable survey strategy, different years (2022, versus the 2018 LGBT National Survey) and different questions (accessing, versus satisfaction with provision) we cannot provide data that speaks to comparison with the general population.

We also note that on reviewing the cited LGBT National Survey for comparator data, we had mis-quoted the figure as 28%, where it was actually 72% found it “not easy” to access mental health care from 24% of people surveyed who had attempted to. We have corrected this in the revised manuscript.

METHODS/DESIGN

The authors refer to a reference from 1975 to support the Delphi approach. Does the authors' approach deviate in any way from this early description of the Delphi approach, and if so, is there evidence to support choice of methods?

Response:

We are grateful for this observation; the proliferation of “Delphi-type” methods since Linstone & Turoff's (1975) landmark book has resulted in adaptations of the method for different applications (notably, in healthcare research). We neglected to cite important examples highlighting this evolution and in particular, the use of online platforms to deliver Delphi studies. We have added a paragraph (lines 199 –222 of the revised manuscript) which highlights the key papers that influenced our design of a Delphi study for the purpose described in the manuscript.

The authors allude to using the snowball recruitment technique in the ethics and dissemination section, however they may wish to consider adding a section in the methods on recruitment of participants. This could include methods they intend to use to recruit not only those with lived experience but also clinicians, ethicists, researchers etc. The authors may also wish to describe the following in this section:

1) reimbursement and compensation being offered to participants (if any),

Response:

Participants will not be financially compensated - see lines 275–278 of the revised manuscript.

2) target recruitment numbers of participants from each stakeholder group,

Response:

We have added recruitment targets (consistent with those in the literature) at lines 233–249 of the revised manuscript.

3) further methods of recruitment that might be used if there is an imbalance in numbers between stakeholder groups,

4) any efforts that will be made to ensure a representative spread of SOGI characteristics within the LGBTQI+ group,

Response (items 3 and 4):

On the issue of ensuring representation over the advisory and survey group - we concluded that explicitly controlling for the diversity and representativeness would be unethical; i.e. it would require us asking potential candidates to submit their SOGI protected/sensitive characteristics and then the executive group 'selecting' members to ensure diversity. Given this, we have proposed to approach different stakeholder groups and instead, we will describe the composition of the advisory (and survey) groups, noting any limitations this imposes on the outputs of the study – please see the “Working Groups: Composition and Recruitment” section in the revised manuscript (lines 279–286)

5) any efforts that will be made to record the geographical spread of participants (even within the UK, it is possible that different regions/nations might have different experiences),

Response:

Similarly to points 3 and 4, we debated collecting information about geography from IP addresses from the online Delphi delivery platform, but felt this was incompatible with our desire to maintain anonymity for the participants (and indeed, a similar issue was raised in our application for research ethics committee approval). We will offer participants the opportunity to self-describe their location, accepting that some may not wish to do so (if, for example, they live in a part of the world with oppressive norms on LGBTQI+ issues). The impact of a Western geographical focus on generalisability can be found in the Discussion, lines 500–513 of the revised manuscript.

6) whether additional participants will be recruited if there are high numbers of drop outs between stages/rounds,

Response:

As the rounds have to be conducted in sequence, iterating on the questionnaire content and for consensus, we require that participants “join” only at the start (round 1) and continue where possible. This prevents “adding” participants to join at later stages in the Delphi (e.g. someone joining in round 2 will not have contributed opinion in round 1 that informs the iterated content presented in round 2). We have described this on lines 240–249 in the revised manuscript.

7) will participants be provided with training.

Response:

Survey group participants will not be provided with training for the Delphi questionnaire rounds; a requirement of participation is that they are able to use a web browser to answer questionnaires. The questionnaires will instruct participants how to complete the forms providing examples.

Literature review stage

This section could benefit from more detail. Whilst it is appreciated that methods within this evidence synthesis stage were perhaps chosen for pragmatic reasons, it could be worth considering using a more systematic approach. As a suggestion, the authors may wish to consider rapid reviews or scoping reviews, depending on the objectives of this stage and the type of literature being reviewed. For example, further details on search strategies, study/publication selection criteria and data extraction might be useful to readers.

Response:

We have refined and added the literature search strategy to the paper; please see the section *Stage 1: Defining the Terms of Reference and Literature review* (lines 296–343) of the revised manuscript, and additionally, our response below. Our literature review will be targeted, rather than systematic or exhaustive, and naturally, it will represent some selection bias of available literature and resources that support our primary aims of developing a toolkit that assists policy makers, healthcare, NGO and engineers/scientists in collecting and reusing existing public assets. We have previously (informally) used public resources on the web e.g. a) construction of gender affirming EHRs, <https://doaskdotell.org/ehr/toolkit/howtoask/>, b) human-computer interaction guidelines <https://www.morgan-klaus.com/gender-guidelines.html> and c) the OECD GEPL policy <https://www.oecd.org/gov/toolkit-for-mainstreaming-and-implementing-gender-equality.pdf>.

These resources will be captured in our targeted review, but might not appear or be indexed as scientific literature.

In addition, each of the areas to be covered in the literature review appear to be separate concepts that could benefit from being considered thoroughly and independently. Will a separate review be conducted for each?

Response:

We acknowledge this would be an important contribution, especially given the distribution of scholarly activity on intersectionality, healthcare and data science that spans a vast range of academic specialisms and literatures. Unfortunately, this same breadth makes it impossible for us to commit the necessary bandwidth for robust systematic or scoping reviews covering these areas (important as they are) as part of this study.

Page 12 – line 11 – Is the creation and review of the Terms of Reference a separate stage requiring its own section in the protocol? How do the authors intend to facilitate the review? Will the participants be given criteria against which they are to critique the Terms of Reference?

Response:

We have revised the section (lines 302–306) of the revised manuscript to detail the ToR stage (briefly: via email, over 4 weeks, the advisory working group will agree the ToR based on an initial document produced by the executive group's literature review). We have also added a Figure to describe the work flow the project overall, which we hope helps to define stages of the study and make clear which groups are involved and responsible for outputs in each stage. While there are no defined criteria

(against which the ToR will be critiqued), the expert opinions of the people in the advisory group will shape the project's concrete focus - for example, which topic areas to cover in the scenarios we present for the re-use of data aim.

Focus group stage?

Other research using a similar approach often includes a focus group stage conducted alongside the literature review in order to fill gaps in the evidence and add depth of understanding. The results of both the literature review and focus groups are then used to inform the first round of the Delphi process. Could the authors comment on their decision to not include focus groups at this stage?

Response:

Our description of the stages and process for the Delphi study were evidently lacking and we have substantially re-organised or re-written these sections of the manuscript. Please see section "Methods/Design", specifically, subsections "Working Groups: Composition and Recruitment", and the subsections describing the four Stages and the accompanying diagram. Please refer to pages 11–20 and the summary contained in the Figure of the revised manuscript.

On the specific point about having a focus group stage: we feel that asking participants to attend a focus group to define the initial questionnaire (for Round 1 of the iterative Delphi rounds) is too burdensome on members of the advisory group. To mitigate the loss of scope and stakeholder input, we have added to Stage 2 a process where via email, the advisory group will be invited to comment on an initial questionnaire design, flowing from the agreed Terms of Reference.

Delphi Stage

How will the Delphi be conducted? In person vs online? Anonymous participants?

Round 1 –Please provide more information on the categorical responses that will be presented to the participants, and how agreement will be judged?

Response:

We neglected to describe the Delphi process in enough detail, and have added a figure to show the division of stages and participants in each stage. We have introduced a "Stage 2" (lines 344–385 of the revised manuscript) which details how responses will be presented and how agreement will be judged. Separately, we have defined a Stage 3 which elaborates on the mechanics (online, pseudonymisation of participants) on lines 386–424 of the revised manuscript.

A minor point is that the stated outcome of the Delphi stage is "consensus statement", however a subsequent stage is described specifically for agreeing the consensus statement. The authors might wish to consider rephrasing the outcome for the Delphi Stage as "draft consensus statement" or similar.

Response:

Thank you for drawing our attention to this error in the original manuscript, and we hope the diagram, revision of the “Stages” (see previous response) and their description clarifies this. Similarly, we have replaced the term “consensus statement” (by explicitly describing outputs) and have made a clearer distinction between Stage 3 (Delphi Rounds), Stage 4 (Consensus Process) and finally, an Outcomes and Dissemination subsection.

Consensus meeting stage

The authors refer to a focus group, however they might wish to avoid using this term as focus group research typically involves techniques that are not described here. Will an experienced facilitator / non-voting chair lead the consensus meeting?

How many participants will you aim for to ensure meaningful engagement?

Will results of previous stage be stratified by stakeholder group prior to being presented?

Response:

We agree and have restructured the Stages subsections and avoided this term. Please refer to subsections Stage 4 and Outcomes and Dissemination which now describe respectively, an online meeting to agree the final statements of agreement over the Delphi questionnaire outputs (chaired and moderated by the executive group) and responsibilities for producing the final outputs of the study.

We think it is difficult to predict how many/which stakeholders (from all groups) will commit to participating in the latter stages of the project so, as for the issue of representation, we will instead report composition of groups at this final meeting in outputs from the study (on the study website and in publications describing the study).

Outcomes/Dissemination Stage

Page 14 Line 50 – how will the results of the prior stages feed in to the white paper? What is your approach to developing the case-study. Is there some text missing from the end of this sentence after “e.g.”?

Response:

We hope that the revised section “Outcomes and Dissemination” now clarifies how the preceding consensus meeting will enable the outputs of the Delph questionnaires to be meaningfully feed into the proposed outputs. We think that the text after “e.g.” should have appeared as three citations to exemplar “playbook” and recommendations guidelines on data stewardship developed by the Ada Lovelace Institute and AI-Now organisations. We are unsure why these did not appear in the original manuscript but it now reads “expect to take the form of a recommendations white paper and

case-study format similar to prior work in related areas [62, 68, 69]" (lines 458–459 of the revised manuscript)

Limitations

A further limitation that the authors may wish to consider – participants may alter their response based to fit the majority view, particularly if there are perceived differences in seniority or expertise.

Response:

We have revised the manuscript to emphasise that the Delphi rounds will be delivered online and anonymously (following the Delphi tradition of mitigating the risk of influential members of a group influencing independent opinions). Please see line 202 (in the description of the Delphi approach), and line 216.

Ethics

Could the authors please consider adding a brief statement on whether ethical approval is required for this study? Is informed consent from participants required? If so, what materials do you intend to provide to facilitate this process?

Response:

We have revised the Ethics and Dissemination section of the manuscript to emphasise that the Delphi rounds will be delivered online and anonymously (following the Delphi tradition of mitigating the risk of influential members of a group influencing independent opinions). Informed consent and ethical approval is required, and the study, participant information/consent materials and design have been approved by the The Institute of Population Health Research Ethics Committee of the University of Liverpool (REC Reference: 12413; 24th July 2023). We have made reference to the consent process in the section "Working Groups: Composition and Recruitment" and as requested, added this detail to the section "Ethics and Dissemination" and the conclusion of the revised manuscript.

Reviewer: 2

Dr. Kyle Harrington, University of Nottingham

Comments to the Author:

The protocol document is of high quality, well-explained and well-motivated. The only two comments I have refer to the "data quality" section of the paper.

Authors suggest there is a misalignment of patient and healthcare professionals expectations around data collection with 80% of healthcare professionals believing they may cause offense compared to 11% of patients reporting likelihood of offence. Given that a healthcare professional may come in to contact with numerous patients, is it really correct to say that the healthcare professionals have misaligned expectations?

Response:

We thank Dr Harrington for this observation – Maragh-Bass *et al* describe a difference between clinician’s expectations (that they will offend by asking SOGI characteristics) and LGBT patient’s reporting they are likely to be offended (80% and 11% respectively). We have re-worded this statement to read “discrepancy between patient and healthcare professionals expectations around offending people by asking about SOGI characteristics, resulting in e.g. 80% healthcare professionals believing they may offend by asking about SOGI characteristics compared to 11% of patients reporting likelihood of offence”

How much does the accessibility of healthcare for LGBTQI+ people differ from the population more generally? Could the authors provide context?

Response:

Another reviewer noted the same point, so we rehearse our response here: We are not aware of any direct comparator data that a) used a similar community sample and b) that speaks directly to the question of difficulty of access.

Various CQC surveys (UK based) report satisfaction results with services (e.g. the 2022 mental community mental health survey <https://www.cqc.org.uk/publications/surveys/community-mental-health-survey>) but only report summaries such as “40% of people had ‘definitely’ seen services enough for their needs”. Given the absence of data with comparable survey strategy, different years (2022, versus the 2018 LGBT National Survey) and different questions (accessing, versus satisfaction with provision) we cannot provide data that speaks to comparison with the general population. We also note that on reviewing the cited LGBT National Survey for comparator data, we had mis-quoted the figure as 28%, where it was actually 72% found it “not easy” to access mental health care from 24% of people surveyed who had attempted to. We have corrected this in the revised manuscript.

VERSION 2 – REVIEW

REVIEWER	Kent, Lisa Queen's University Belfast, Centre for Public Health
REVIEW RETURNED	19-Feb-2024
GENERAL COMMENTS	My thanks to the authors for their considered response to suggestions and queries raised. The manuscript reads well and is enhanced by the addition of Figure 1 which describes the stages of the Delphi study. I wish the group the best of luck for their study.

VERSION 2 – AUTHOR RESPONSE